# Effect of Photolithographic Biomimetic Surface Microstructure on Wettability and Droplet Evaporation Process

**DOI:** 10.3390/biomimetics9120724

**Published:** 2024-11-24

**Authors:** Zhihao Zhang, Yuying Yan

**Affiliations:** Faculty of Engineering, University of Nottingham, University Park, Nottingham NG7 2RD, UK; zhihao.zhang1@nottingham.ac.uk

**Keywords:** photoresist, bionic surface, droplet evaporation, wetting phenomena

## Abstract

In nature, engineering technology and daily life, wetting phenomena are widespread and have essential roles and significance. Bionics is becoming increasingly important nowadays and exploring the mechanism that influences biomimetic surface microstructure on droplet wetting process and heat and mass transfer characteristics is becoming more meaningful. In this paper, based on photolithography technology, SU-8 photoresist was used as raw material to prepare biomimetic surfaces with microstructures in various arrangements. The research results show that the wettability of biomimetic functional surfaces can be regulated by regulating the shape and arrangement of photoresist micro-pillars. At the same time, the effects of surface microstructure configuration and roughness on the heat and mass transfer processes within the droplets were also comprehensively studied. The results show that a biomimetic surface with cylindrical micro-pillars can effectively inhibit the evaporative cooling effect of the liquid–vapour interface. This effect becomes more evident with the increase in roughness, and the interface temperature difference can be reduced by up to 18%. Similarly, the biomimetic surface with cylindrical micro-pillars can also effectively promote the evaporation rate of sessile droplets, which can be increased by about 13%. In addition, the research also shows that regardless of the structure, substrate temperature changes will significantly impact the wetting phenomenon of the biomimetic surface. This study aims to guide the optimal design of biomimetic surfaces prepared based on photoresistance.

## 1. Introduction

Bionic principles, as an independent discipline born in the 1950s and 1960s [1], have been widely used in the entire field of science and engineering [2,3,4]. Among them, functional biomimetic surfaces can give a surface the unique functions it requires by simulating the various microstructures, physics, chemistry and other characteristics of the surface or skin of plants and animals, giving them broad application prospects [5,6]. Based on this, the design and preparation of biomimetic functional surfaces have received more and more attention [7,8]. Particularly due to the development of the electronics industry, biomimetic surfaces have been further combined and applied in emerging electronic equipment fields such as flexible devices [9], biosensors [10] and electronic skin [11] due to their unique properties. As the cornerstone of the electronic manufacturing field, the photolithography process determines the performance of the advanced electronic equipment and products mentioned above [12,13]. Therefore, applying photolithography technology to prepare biomimetic surfaces is significant to the bionic design of electronic equipment.

Regarding the preparation of biomimetic functional surfaces, whether using photolithography or other means, the design of biomimetic surface microstructures is always crucial, and an essential part of this is the choice of a micro- or nanostructure configuration [14,15]. Wu et al. [16], based on electro-spun nanofibers, fabricated a series of natural biological bio-inspired surfaces exhibiting anisotropic wettability in two to three directions. Kostal et al. [17] combined hydrophilic and super-hydrophobic microstructures on a biomimetic surface to form a composite structure. The results show that a biomimetic surface with this type of pattern can achieve a 60% increase in fog collection efficiency. Similarly, based on fog collection, Azad et al. [18] and Choi et al. [19] have produced bionic functional surfaces with super-hydrophilic and dual wettability microstructures, improving their fog collection efficiency. Based on natural inspiration, Jian et al. [20] prepared a biomimetic surface with a micron-nanoscale hierarchical structure based on nano-fumed silica, which has significant super-hydrophobic properties and excellent anti-icing and anti-frost functions. Xu et al. [21] prepared a hierarchical micro/nano super-hydrophobic surface on the surface of a copper metal foam through a solution immersion method, which had a high mechanical stability. Bio-inspired by the bionics of hydrophobic springtail skin, Yin et al. [22] prepared a biomimetic super-hydrophobic surface with a mushroom-shaped microstructure through 3D lithography printing. They proved that the realisation of hydrophobicity has nothing to do with the material and is entirely due to geometric factors. Baron et al. [23] prepared a bionic surface with a structure similar to the skin of reptiles and insects through femtosecond laser pulse processing, which can achieve various wettability performances from hydrophilic to super-hydrophobic. Stratakis et al. [24] also used pulsed laser processing to process micro/nano bionic structures on the silicon surface to obtain different degrees of wettability. Similarly, Huang et al. [25] prepared micro-silicon pillar substrates by photolithography and studied the effects of roughness and substrate temperature on the evaporation characteristics of sessile droplets. The dot/line microstructure patterns prepared by Zhang et al. [26] can control the surface adhesion strength and wettability direction, which is reversible. Lu et al. [27] also used laser technology, bio-inspired by the leaves of Bauhinia, to etch circular patterns on the surface of stainless steel, thus preparing a bionic super-hydrophobic surface. Wei et al. [28] used laser technology to etch micropapillary structures on the magnesium alloy surface and prepared biomimetic super-hydrophobic surfaces. Li et al. [29] prepared a honeycomb bionic hierarchical structure on the aluminium alloy surface through laser ablation technology, achieving an excellent super-hydrophobic effect. Liang et al. [30] used high-speed wire-cutting technology to prepare a bionic functional surface inspired by cicada wings on the surface of aluminium alloy, which also has good super-hydrophobic properties. As mentioned above, extensive and in-depth research has been conducted on the configuration of microstructures. Still, most of the research is based on achieving the super-hydrophobic properties of the biomimetic surface [31], and research based on conventional hydrophilic biomimetic surfaces or the transformation of surface wettability is still relatively rare.

In the preparation process for biomimetic functional surfaces, in addition to the design mentioned above of the surface configuration, choosing the material used to fabricate the microstructure also has a crucial influence on the overall performance of the biomimetic surface [32,33]. At present, profiting from the diversity of non-metallic materials, the preparation of bionic surfaces based on them is receiving widespread attention [34]. Hu et al. [35] used photosensitive resin as a raw material and adopted a 3D printing method to prepare a spring bionic surface with an improved waterproof performance. Liu et al. [36] also used 3D printing to prepare a super-hydrophobic biomimetic surface inspired by Nepenthes peristome, with a petal-like microstructure using fluid resin. Based on femtosecond laser technology, Bian et al. [37] used PDMS and silicone as raw materials to prepare the biomimetic functional surface bio-inspired by lotus leaves, giving it anisotropy of interfacial resistance for droplet rolling. In addition, being bio-inspired by the cactus, Wang et al. [38] prepared a bump array arrangement on a silicon wafer through an etching process. Then, they subjected its surface to super-hydrophobic treatment to prepare a super-hydrophobic surface with an impressive water-collecting function. At the same time, the preparation of biomimetic surfaces based on metal materials has also been widely studied [39,40]. Being bio-inspired by the natural surface of rice leaves, Yang et al. [41] prepared groove and nipple bionic microstructures on an aluminium alloy surface, achieving an excellent hydrophobic effect. Wan et al. [42] also prepared an anti-condensation biomimetic functional surface with a micro-groove structure based on an aluminium alloy surface, naturally bio-inspired by the surface of a bamboo leaf. Similarly, inspired by the nature of reed leaves, Gao et al. [43] prepared a biomimetic functional surface with anisotropic and isotropic switchable wettability on the copper metal surface. In addition, the preparation of bionic functional surfaces based on various other materials has also received widespread attention. Inspired by various organisms, Feng et al. [44] successfully prepared a patterned smooth super-hydrophilic surface based on ink direct printing, significantly improving the water collection efficiency by about 139%. Yao et al. [45] prepared a lotus-shaped bionic film based on fluorine-containing polymers with excellent super-hydrophobic properties. So far, in the research on biomimetic functional surfaces, numerous materials have been selected in the preparation process to achieve various properties. However, in these studies, research on biomimetic functional surfaces directly based on photoresist materials is still rare, and the influence of photoresist material characteristics on the wetting properties of biomimetic surface microstructures remains to be studied.

In summary, due to its excellent application value, research on bionic surface microstructures has been widely conducted. However, most of the research is based on the realisation of super-hydrophobic properties, and there are few studies on hydrophilic biomimetic surfaces or the hydrophilic–hydrophobic transformation of biomimetic surfaces. Moreover, most preparations of bionic surfaces are based on common metal or non-metal materials, and there are still few studies on directly preparing surface bionic microstructures based on using photoresists as raw materials. Therefore, this paper uses the SU-8 photoresist to prepare biomimetic functional surfaces with various microstructures on silicon substrates based on bio-inspiration. The effects of microstructure parameters on the wettability of photoresist biomimetic surfaces and their sessile droplet heat and mass transfer characteristics are studied using deionised water as the working fluid. Section 1 and Section 2 introduce the research status and experimental preparation. In Section 3 of this paper, the effects of micropillar shape, micropillar arrangement (roughness) and substrate temperature change on the wettability of the biomimetic surface and the heat and mass transfer process at the droplet’s liquid–vapour interface are studied. In Section 4, the research results are summarised. This work explores the influence of biomimetic surface microstructure design on wettability and provides effective properties of biomimetic microstructures in photoresists.

## 2. Material and Experimental Setup

The bionic functional surface in this study was designed based on the concept of bionics, which was proposed in 1958, and the term “Biomimetics” was introduced in 1991 [46]. As shown in Figure 1a, plant surfaces and animal skin in nature have some unique characteristics, and thus, based on observing the microstructure of their surfaces, bionic microstructures can be designed. Among a series of bionic microstructures, etching a micro-pillar on the surface is one of the most widely used design strategies [47,48]. Based on this, as shown in Figure 1b, this study also used SU-8 as a raw material to prepare micro-pillar structures on the silicon wafer to realise the preparation of biomimetic functional surfaces. The morphology of the prepared biomimetic functional surface is shown in Figure 1c. Some physical properties of the photoresist SU-8 2000 series are shown in Table 1.

The specific process of preparing the micro-pillar structure on the silicon wafer surface is shown in Figure 2. As shown in the figure, the substrate was first cleaned with acetone solvent, and after cleaning, it was baked at 200 °C for 30 min to remove surface water molecules. After that, the SU-8 photoresist was dispersed and spin-coated onto the substrate surface using a spinner. It is worth mentioning that the SU-8 photoresist has a wide range of applications in micro–nano devices, electronic technology and even biomedical fields, thanks to its high resolution and stable thermo-electromechanical and chemical properties [49,50,51]. Therefore, based on its wide range of applications and potential, SU-8 was selected as the material for directly preparing surface bionic microstructures in this paper. Then, the photoresist was soft-baked and slowly cooled. After that, the photoresist was exposed to ultraviolet light through a mask reticle to prepare the desired pattern and a solvent was used to remove the unexposed areas. Then, the final stage was to hard-bake the photoresist at 150–200 °C to permanently solidify it, ultimately obtaining the microstructure required for the design. Meanwhile, it also can be seen in Figure 2 that the parameters that determine the configuration and arrangement of the prepared micro-pillars are mainly the pillar spacing *S*, the pillar height *H* and the pillar diameter (cylindrical pillar)/side length (square pillar) *D*. The configuration parameters of each surface are shown in Table 2. It is worth mentioning that, considering the processing accuracy of SU-8 photoresist (0.5 to 200 µm), the processing structure parameters were selected between 10 and 50 microns. At the same time, based on various configuration parameters, the roughness factor on the surface can also be calculated based on Equations (1) and (2), and the result is also listed in Table 2.

In addition, the experimental equipment and process used in the study of sessile droplet evaporation are also shown in Figure 3. The droplets can be stably placed through the pipette (London LaboQuip, London, UK) on the biomimetic functional surface, which has a heating device at the bottom to control the bottom surface temperature between 50 and 80 °C. During the evaporation process of the droplet, the droplet profile, such as the change in its contact angle and height, is measured in real-time by the Optical Tensiometers (Biolin Scientific, Espoo, Finland). At the same time, during the evaporation process, the temperature distribution at the droplet liquid-vapour interface is also recorded by the IR Camera (FLIR LLC, Wilsonville, OR, USA). At the same time, as the bottom temperature changes, the temperature of the droplet will also change, and its thermophysical properties will also change. The changes in the thermophysical parameters of deionized (DI) water under different temperature conditions are shown in Table 3.

In addition, there are different calculation methods for the roughness factor for square and cylindrical pillars. When the surface microstructure is a square pillar, the roughness factor can be calculated by Equation (1):(1)$$f=\frac{{\left(S+L\right)}^{2}+4\cdot L\cdot H}{{\left(S+L\right)}^{2}}$$

In addition, when the surface microstructure is a cylindrical pillar, the roughness factor can be calculated by Equation (2):(2)$$f=\frac{{\left(S+D\right)}^{2}+\pi \cdot D\cdot H}{{\left(S+D\right)}^{2}}$$

At the same time, the Wenzel model could be expressed as:(3)$$cos\theta^{*}=r\cdot cos\theta$$
where *θ** is the measured contact angle, *θ* is the contact angle with a smooth surface, and *r* is the surface roughness,

In addition, the Cassie–Baxter equation could be expressed as:(4)$$cos\theta^{*}=r\cdot f_{SL}\cdot cos\theta +f_{SL}-1$$
where the *f_SL_* is the proportion of the actual wetted surface.

## 3. Results and Discussion

### 3.1. Effect of Microstructure Configuration on Biomimetic Surface Wettability

This study first investigated the effect of micro-pillar shape on the performance of biomimetic functional surfaces. First, two surfaces, SS-1 and CS-3, were selected, whose surface microstructures have similar dimensions (pillar height: 25 µm; pillar spacing: 20 µm; pillar side length/diameter: 30 µm). The changes in the contact angle and dimensionless contact diameter during the evaporation of a 1.0 μL DI water droplet on the surface are shown in Figure 4, where *d*_0_ is the initial contact diameter, and *d* is the real-time contact diameter during the evaporation process. The evaporation process of the droplet’s evaporation when the substrate temperature is 50 °C is shown in Figure 4a. The contact angle of the droplet on the SS-1 surface is more significant than that of the droplet on the CS-3 surface, with a difference of about 9%. It is also worth noting that the contact angle on the SS-1 surface is greater than 90°, which is hydrophobic, while the contact angle on the CS-3 surface is less than 90°, which is hydrophilic. As evaporation proceeds, the contact angle gradually decreases. At the end of evaporation, since the measurement of the contact angle of the droplet becomes difficult, the change in the contact angle of the droplet is represented by a dotted line. Therefore, in this paper, since the last stage of evaporation occupies a very short period of the entire process and has a minimal impact on the overall process, it is ignored to simplify the calculation and analysis. At the same time, the contact line remains fixed on the SS-1 or CS-3 surface in the early stage of droplet evaporation. Still, as the evaporation proceeds, the contact line of the droplet begins to shrink, and the contact diameter decreases by about 15%. Therefore, the droplet experiences two evaporation modes on the biomimetic surface: the constant contact radius (CCR) mode and the Mixed mode. The CCR mode refers to the contact radius remaining unchanged. In contrast, the Mixed mode refers to the contact radius and contact angle decreasing simultaneously during evaporation. In addition, the process of the change in the droplet when the substrate temperature increases from 50 °C to 60 °C, 70 °C or 80 °C is shown in Figure 4b, Figure 4c and Figure 4d, respectively. The total evaporation life of the droplets decreases with the increase in temperature, and the evaporation mode of the droplets still maintains the two modes of CCR and Mixed.

At the same time, the changes in the dimensionless height of the droplet on CS-3 and SS-1 under different substrate temperature conditions are shown in Figure 5a, where *h*_0_ is the initial height of the droplet and *h* is the real-time height of the droplet during the evaporation process. With the increase in temperature, the rate of change in droplet height gradually increases, increasing by about 4 times from 50 °C to 80 °C, which is caused by the significant acceleration of the evaporation rate. At the same time, the height reduction rate of the droplet on the CS-3 surface is always slightly higher than that on the SS-1 surface, which is due to the stronger wettability of the CS-3 surface. In addition, the changes in the droplet’s initial equilibrium contact angle (ECA) under different surface and substrate temperatures are shown in Figure 5b. On both the CS-3 and SS-1 surfaces, the contact angle of the DI water droplet decreases with the increase in the substrate temperature, and the average decrease rates are −0.19°/°C and −0.18°/°C, respectively. In addition, the droplets on the CS-3 surface are always in a hydrophilic state (ECA < 90°), while the droplets on the SS-1 surface are basically in a hydrophobic state (ECA > 90°). This phenomenon can be attributed to the fact that the droplets on the CS-3 surface are in the Wenzel state, while the droplets on the SS-1 surface are in the Cassie–Baxter state.

### 3.2. Effect of Microstructure Arrangement on Biomimetic Surface Wettability

The previous Section studied the influence of the shape of the micropillar structure on the wettability of biomimetic surfaces. In this section, we will comprehensively explore the effects of the configuration and arrangement of micro-structures on the wettability of biomimetic surfaces. The evaporation process of the droplet on the different surfaces when the substrate temperature is 50 °C, 60 °C, 70 °C and 80 °C is shown in Figure 6a, Figure 6b, Figure 6c and Figure 6d, respectively. Furthermore, the contact angle of the droplet on the SS surface is always greater than that on the CS series surface, which indicates that the wettability of the CS series surface is always more substantial than that of the SS series surface. In addition, as the evaporation proceeds, the contact angle of the droplet continuously decreases, while the contact line is in a pinned state in the first half and then shrinks as the evaporation proceeds. Therefore, no matter which surface is being used, the droplet evaporation process will transition from the CCR to the Mixed mode. In addition, it can be seen from Figure 6b–d that when the substrate temperature increases, its evaporation process does not change significantly, except that the lifetime is significantly reduced. The droplet’s evaporation mode still transitions from the CCR to the Mixed mode. The wettability of the CS series surface is always more substantial than that of the SS series surface. In addition, it can also be found that the SS series surfaces maintain the droplet pinning state for a longer time, contributing to its ability to produce a hydrophobic state.

In summary, the size and arrangement of biomimetic surface microstructures significantly impact wettability. For the purposes of comparison, the magnitudes of the initial equilibrium contact angles of the droplets on the CS and SS series surfaces are shown in Figure 7a. The ECA of the droplets on the SS series surface is greater than 90°, and they are all hydrophobic. However, the ECA of the droplets on the CS-1 surface is greater than 90° at low temperatures (*T*_sub_ < 70 °C), and the ECA gradually decreases and is less than 90° as the temperature rises to 80 °C. This phenomenon might be attributed to the change in the droplet contact angle from the Cassie–Baxter state to the Wenzel state. In addition, on the CS-2 and CS-3 surfaces, the ECA of the droplets was consistently below 90°, indicating their hydrophilicity. Moreover, SS series surfaces can increase the contact angle by about 12% on average compared with CS series surfaces. This could be attributed to the square pillar microstructure having denser gaps, making it easier for droplets to be in the Cassie–Baxter wetting state. Therefore, the cylindrical micro-pillars can achieve the hydrophobic properties ofthe biomimetic surface under certain conditions. In contrast, this property is more accessible using a square micro-pillar structure. At the same time, on any surface, the ECA of DI water droplets decreases with increasing temperature, and the average rate of decrease is about 0.17°/°C. At the same time, under different substrate temperatures and surfaces, the changes in the work of adhesion (WoA) at the solid–liquid interface at the initial equilibrium state of the droplet are shown in Figure 7b. The work of adhesion on the CS series surface is always greater than that on the SS surface, with the maximum difference reaching up to 36%. This means that more work is required to separate the droplet on the CS series surface or that more energy is released during the wetting process of the droplet on the CS series surface.

### 3.3. Effect of Microstructure Arrangement on Sessile Droplet’s Liquid–Vapour Interface Heat Transfer

In addition to the wetting characteristics, this article will continue by exploring the influence of biomimetic functional surface microstructure on the sessile droplet interfacial heat transfer characteristics. Taking the CS-3 and SS-3 surfaces as examples, when the substrate temperatures are 50 °C and 80 °C, the temperature distribution changes at the droplet’s liquid–vapour interface when the substrate temperatures are 50 °C and 80 °C are shown in Figure 8a and Figure 8b, respectively. It can be seen, intuitively, that no matter if the substrate temperature increases from 50 °C to 80 °C, the temperature distribution of the liquid–vapour interface of the droplet on the CS-3 surface will be more uniform each period. In addition, as the evaporation process proceeds, the temperature of the droplet’s liquid–vapour interface gradually increases. These phenomena can be mainly attributed to the fact that, as evaporation proceeds, the height of the droplets gradually decreases, which enhances the heat transfer inside the droplets and thus gradually increases the temperature of the liquid–vapour interface.

In addition, it can be seen from Figure 8 that the temperature distribution of the liquid–vapour interface of the droplets on the SS-3 surface is more uneven at almost every stage of droplet evaporation. At the same time, the temperature distribution at the central line position of the droplet at the beginning, middle and end of evaporation under the substrate temperature range from 50 to 80 °C is shown in Figure 9. Among them, the temperature distribution when the droplet is located on the CS-3 surface when the substrate temperature increases from 50 °C to 80 °C is shown in Figure 9a, Figure 9c, Figure 9e and Figure 9g, respectively. As the substrate temperature increases, the non-uniformity of the temperature of the droplets in the initial stage also increases, up to 37%. This phenomenon also occurs on the SS-3 surface, as shown in Figure 9b,d,f,h. In addition, by comparison, it can be found that, under the same substrate temperature conditions, the temperature distribution of the droplet’s liquid–vapour interface on the SS-3 surface is more uneven, which could be attributed to the hydrophobicity of the SS series surface.

In addition, the droplet’s liquid–vapour interface’s maximum temperature difference (*T*_diff, max_) under various surface and substrate temperature conditions is shown in Figure 10a. The maximum temperature difference refers to the temperature difference between the centre of the droplet’s liquid–vapour interface and the contact line, such as the mark in Figure 8. On any surface, the maximum temperature difference in the droplets always increases with increasing substrate temperature, up to a difference of 37%. At the same time, when the substrate temperature increases, the gap in the maximum temperature difference between different surfaces remains unchanged basically, such as the maximum temperature difference between the SS-3 and CS-3 surfaces kept at 2.5 °C when the substrate temperature rises from 50 °C to 80 °C. At the same time, when the substrate temperature is in the range of 50–80 °C, the CS surface can reduce the maximum temperature difference by about 18% compared with the SS surface. These phenomena indicate that the evaporative cooling effect at the droplet’s liquid–vapour interface will be enhanced as the biomimetic surface hydrophobicity increases. In addition, the changing trends in the average evaporation rate of droplets on various surfaces under different substrate temperature conditions are also shown in Figure 10b. As the temperature increases, the evaporation rate of the droplets increases significantly, increasing by about 4 times on average from 50 °C to 80 °C. In addition, the evaporation rate of the droplets on the SS series surface is always lower than that on the CS series surface, with an average decrease of about 13%, which can be attributed to the dual effects of decreased internal heat transfer of the droplets and enhanced interfacial evaporative cooling effect.

## 4. Conclusions

Inspired by natural bionics, this study designed a bionic functional surface with a micro-pillar structure through photolithography technology. Based on this, the effects of the microstructure structural parameters and arrangement on the biomimetic surface wettability and the heat and mass transfer process at the liquid–vapour interface of the sessile droplets were further studied. The research results show that under conditions of the same or similar size parameters, the arrangement and structural changes of the photoresist bionic micro-pillars will change the gap tightness and roughness factor, which may affect the surface wettability. In addition, the substrate temperature will also affect the wettability. As the temperature increases, the contact angle of the DI water droplets on any surface will decrease, enhancing the biomimetic surface wettability. In addition, the microstructure of the biomimetic surface also significantly impacts the heat and mass transfer process at the liquid–vapour interface of the sessile droplets. Studies have shown that, compared with the CS series surfaces, the liquid–vapour interface of the droplets on the SS series surfaces has a more uneven temperature distribution, with an average difference of up to 18%. This can be attributed to the fact that the SS series surface not only increases the thermal resistance inside the droplet but also causes a more obvious evaporative cooling effect at the liquid-vapour interface. Naturally, under the combined effect of these factors, the evaporation rate of droplets on the SS series surface is suppressed, and the reduction can reach 13% compared with the CS series surface. In addition, this paper prepared a biomimetic functional surface based on SU-8 as the raw material. This study shows the feasibility of using SU-8 directly to prepare the required functional surface with weak hydrophobic and weak hydrophilic properties. This research method is expected to guide the use of SU-8 in the preparation of bionic equipment in the fields of biomedicine, semiconductors, electronic packaging, solar energy and even micro fuel cells.

## Figures and Tables

**Figure 1 biomimetics-09-00724-f001:**
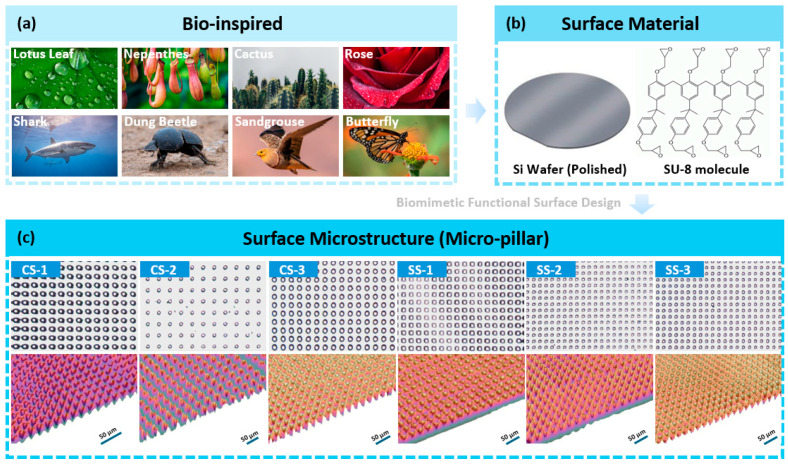
(**a**) Common bionic prototypes in the biomimetic surface design process (**b**) raw materials used for surface preparation; and (**c**) schematic diagram of the three-dimensional structure of photoresist micropillars.

**Figure 2 biomimetics-09-00724-f002:**
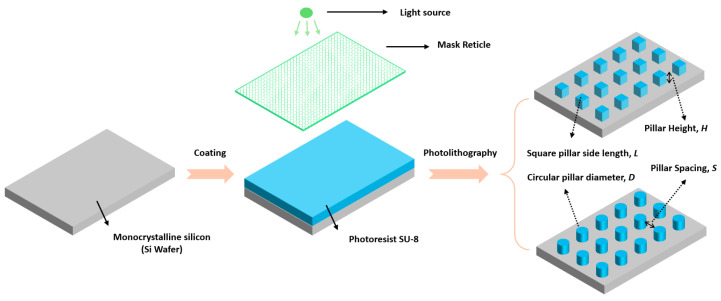
Photoresist biomimetic surface production process.

**Figure 3 biomimetics-09-00724-f003:**
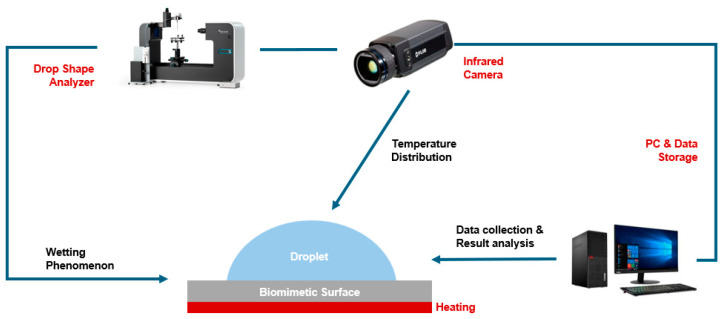
Equipment and operation procedures for the evaporation experiment of sessile droplets on bionic surfaces.

**Figure 4 biomimetics-09-00724-f004:**
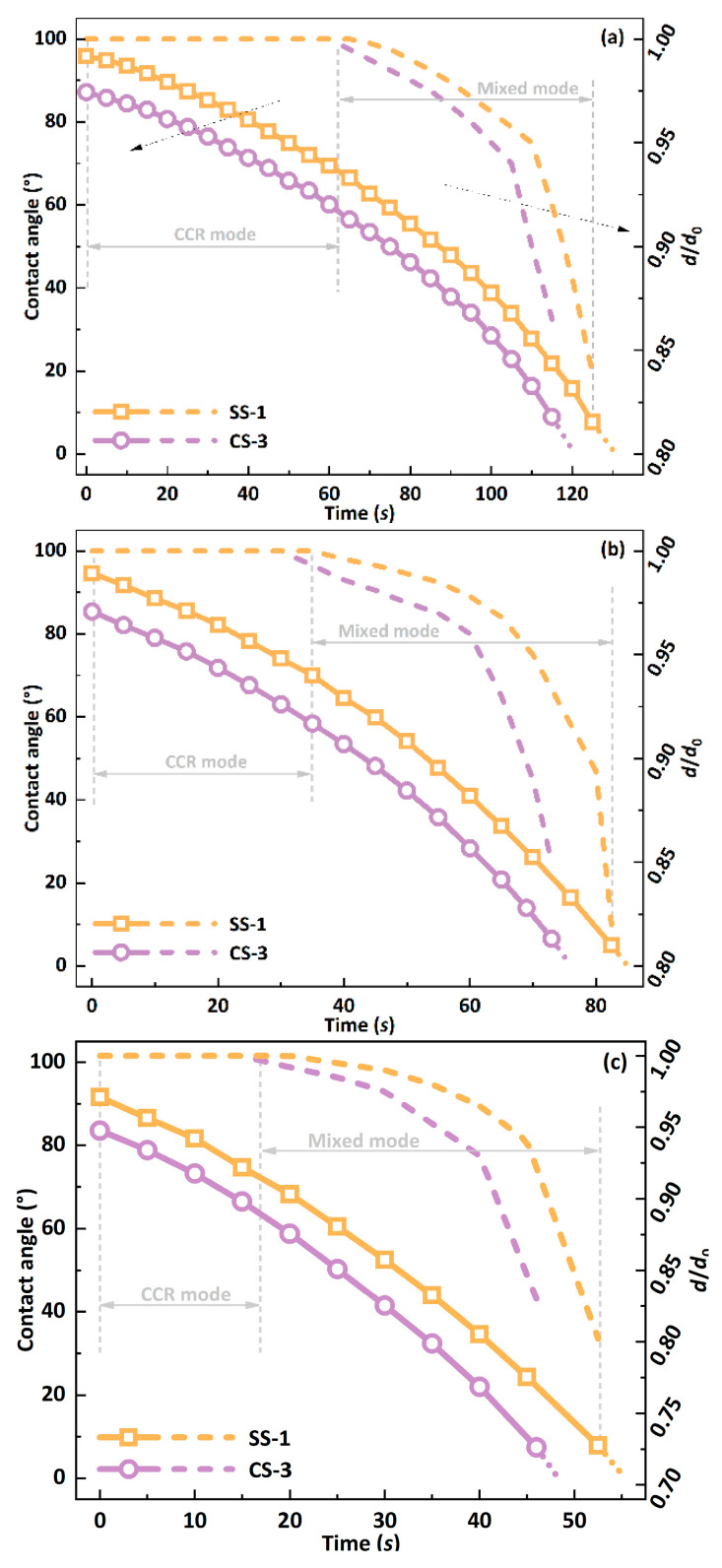
Changes in contact angle and contact line of a 1.0 μL droplet on SS-1 and CS-3 surfaces at a substrate temperature of (**a**) 50 °C; (**b**) 60 °C; (**c**) 70 °C; (**d**) 80 °C.

**Figure 5 biomimetics-09-00724-f005:**
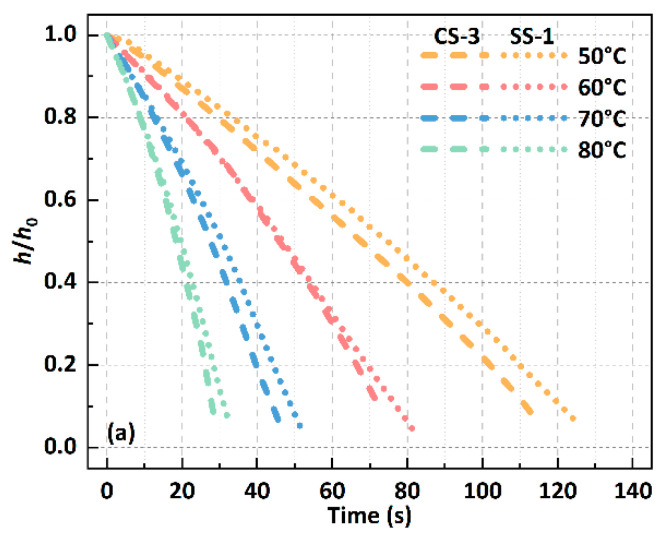
(**a**) Dimensionless height of a 1.0 μL droplet over time for different surfaces and substrate temperatures; (**b**) initial equilibrium contact angle of a 1.0 μL droplet on different surfaces and substrate temperatures.

**Figure 6 biomimetics-09-00724-f006:**
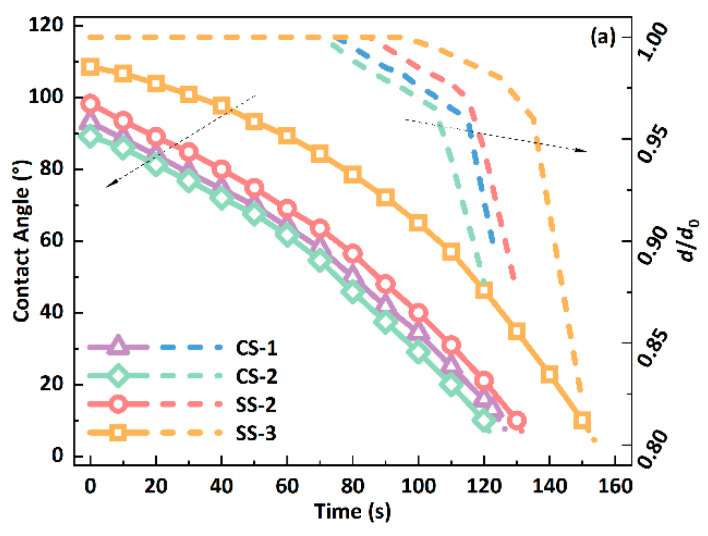
Changes in contact angle and contact line of a 1.0 μL droplet on different surfaces at a substrate temperature of (**a**) 50 °C; (**b**) 60 °C; (**c**) 70 °C; (**d**) 80 °C.

**Figure 7 biomimetics-09-00724-f007:**
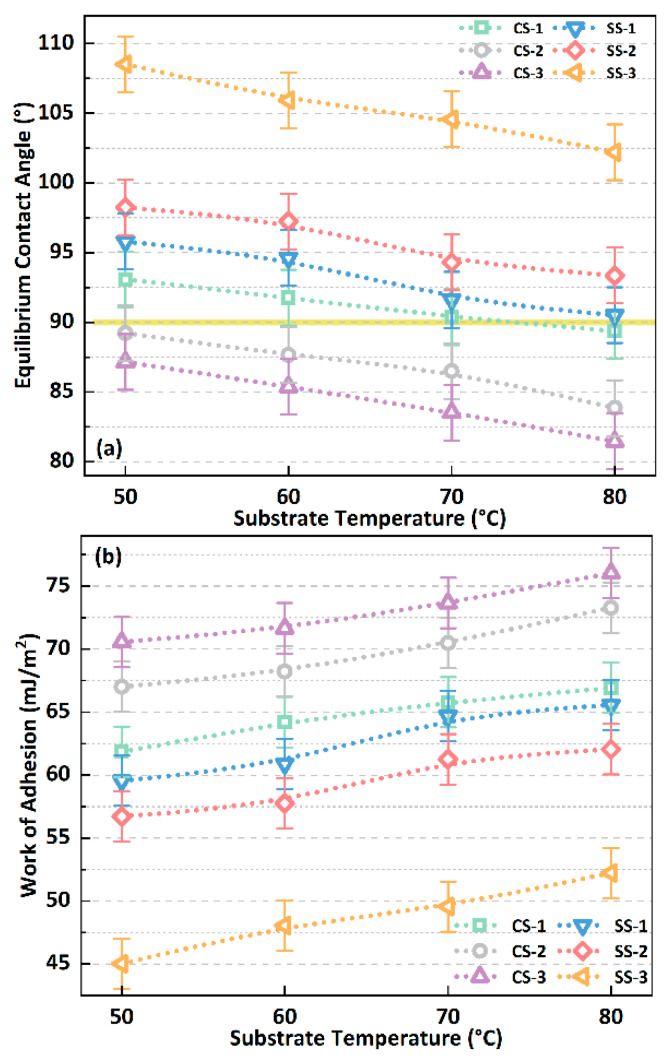
(**a**) The initial equilibrium contact angle of the droplet changes under different surface and substrate temperature conditions; (**b**) the initial adhesion work of the droplet changes under different surface and substrate temperature conditions.

**Figure 8 biomimetics-09-00724-f008:**
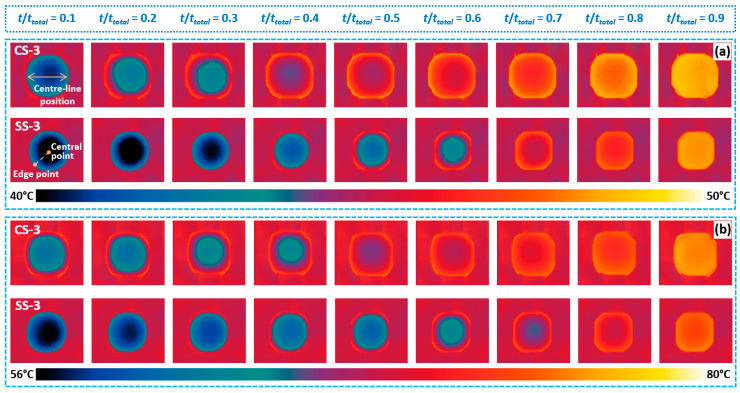
Temperature distribution in the droplet liquid–vapour interface on the CS-3 and SS-3 surfaces at different periods when the substrate temperature is (**a**) 50 °C and (**b**) 80 °C.

**Figure 9 biomimetics-09-00724-f009:**
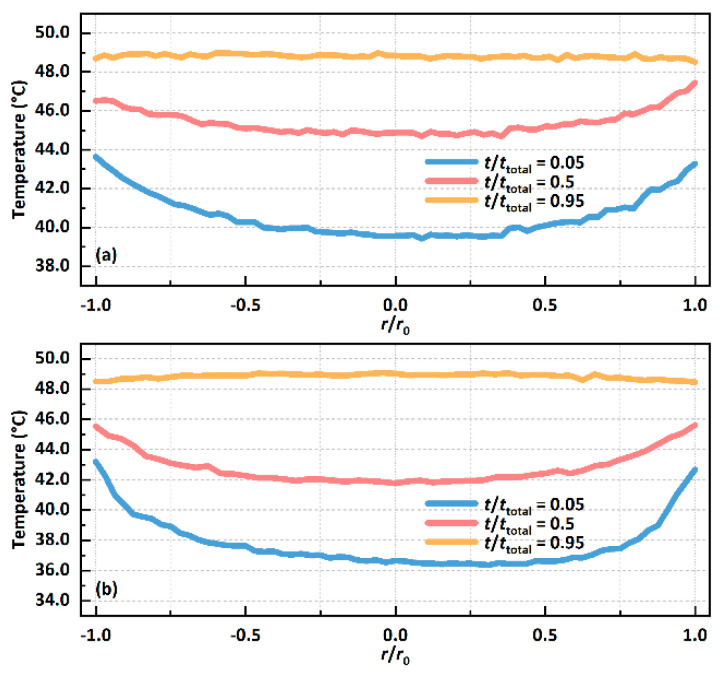
The temperature distribution at the centre-line of the droplet’s liquid–vapour interface at different periods when the substrate temperature of the CS-3 surface is (**a**) 50 °C, (**c**) 60 °C, (**e**) 70 °C or (**g**) 80 °C; the temperature distribution at the centre line of the droplet’s liquid–vapour interface at different periods when the substrate temperature of the SS-3 surface is (**b**) 50 °C, (**d**) 60 °C, (**f**) 70 °C or (**h**) 80 °C.

**Figure 10 biomimetics-09-00724-f010:**
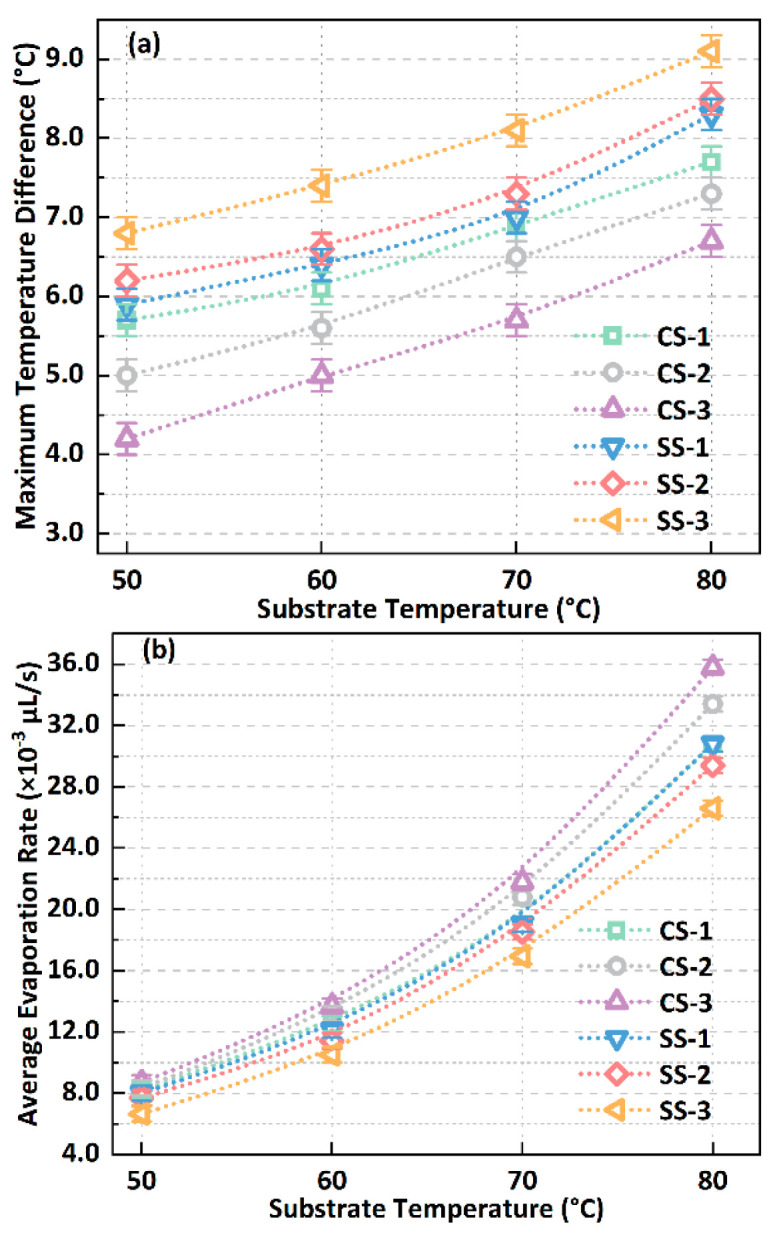
(**a**) Temperature difference at the droplet’s liquid–vapour interface in the initial state under different surface and substrate temperature conditions; (**b**) variation in the overall average evaporation rate of the droplet with the surface and substrate temperature.

**Table 1 biomimetics-09-00724-t001:** Basic physical properties of photoresist SU-8 2010.

Physical Properties	Approximate Values
Glass Transition Temperature (°C)	210
Thermal Stability (°C @ 5% wt. loss)	315
Thermal Conductivity (W/mK)	0.3
Coeff. of Thermal Expansion (ppm)	52
Adhesion Strength (mPa)	38

**Table 2 biomimetics-09-00724-t002:** Biomimetic surface microstructural parameters.

	CS-1	CS-2	CS-3	SS-1	SS-2	SS-3
Average height *H* (µm)	15	50	30	30	30	40
Diameter/Side Length *D* (µm)	20	20	30	30	20	20
Spacing *S* (µm)	15	40	20	20	20	20
Roughness factor (*f*)	1.769	1.873	2.131	2.44	2.50	3.0

**Table 3 biomimetics-09-00724-t003:** Physical properties of deionized water at different temperatures.

	50 °C	60 °C	70 °C	80 °C
Density (g/cm^3^)	0.98804	0.98321	0.97778	0.97180
Viscosity (mPa·s)	0.5494	0.4688	0.4061	0.3635
Surface Tension (dyn/cm)	67.91	66.18	64.42	62.61
Thermal Conductivity (mW/m·K)	640.60	650.91	659.69	667.02

## Data Availability

The authors will supply the relevant data in response to reasonable requests.

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
