# Peer review of "Effect of Photolithographic Biomimetic Surface Microstructure on Wettability and Droplet Evaporation Process"

_biomimetics, 2024, doi:10.3390/biomimetics9120724_

Round 1

Reviewer 1 Report

Comments and Suggestions for Authors

The article studies the effect of photolithographic biomimetic surface microstructure on and droplet evaporation process. The article has certain research value. But there are two questions that the author should consider.

1. The use of SU-8 photoresist as a raw material to prepare a bionic surface with a microstructure is mentioned, but there is no detailed explanation of why SU-8 photoresist is selected, as well as the advantages and challenges of this material in the preparation of bionic surfaces. Recommended addition.

2. It is suggested that the authors emphasize in the conclusion part how their findings promote research in the field of bionic surfaces and the potential impact of these findings on future design and applications.

Author Response

Reviewer-1 comments

The article studies the effect of photolithographic biomimetic surface microstructure on and droplet evaporation process. The article has certain research value. But there are two questions that the author should consider.

  1. The use of SU-8 photoresist as a raw material to prepare a bionic surface with a microstructure is mentioned, but there is no detailed explanation of why SU-8 photoresist is selected, as well as the advantages and challenges of this material in the preparation of bionic surfaces. Recommended addition.

Response:  Thanks to the reviewer for the valuable suggestions. Section 2 now includes the introduction, significance, and advantages of SU-8 material, which are marked with a yellow background.

2. It is suggested that the authors emphasize in the conclusion part how their findings promote research in the field of bionic surfaces and the potential impact of these findings on future design and applications.

Response: Thank the reviewer for the positive comments and valuable suggestions for improving the quality of our manuscript. The research and application significance of this research result in the field of bionics has been added to the end and marked with a yellow background.

Reviewer 2 Report

Comments and Suggestions for Authors

biomimetics-3303949

In the manuscript with the title “Effect of photolithographic biomimetic surface microstructure on wettability and droplet evaporation process” the authors investigate how biomimetic surface microstructures, created using photolithography, influence wettability and droplet evaporation processes. By experimenting with various microstructures, such as square and cylindrical micro-pillars, the authors found that square micro-pillars tend to enhance hydrophobicity more effectively. In contrast, cylindrical micro-pillars reduce the temperature difference at the liquid-vapor interface by up to 18% compared to square micro-pillars, thereby mitigating evaporative cooling, and enhance droplet evaporation rates by approximately 13% compared to surfaces with square micro-pillars. Increased roughness amplifies these effects, while substrate temperature changes strongly influence surface wetting. These insights aim to support the optimal design of biomimetic surfaces. This research advances the understanding of how biomimetic surface microstructures impact wettability and evaporation dynamics, offering valuable insights for designing optimized surfaces in fields such as biomimetics, surface engineering, and thermal management.

The manuscript has some limitations and weaknesses that need to be addressed and improved. Here are some specific recommendations and suggestions for each section of the manuscript:

Materials and Experimental setup

1.     To enhance clarity in Figure 1, the authors should add a clear label to each image with an index (e.g., 'a,' 'b,' 'c') and include a description of what each image represents within the figure caption.

2.     While the authors reference the general strategy of using micro-pillars ([46, 47]), there is no explicit linkage between these references and the specific choices of processing parameters. Although the manuscript provides a description of the processing parameters and test conditions, it would benefit from a more explicit discussion on how these choices were influenced by prior studies or preliminary experiments.

Results and discussion

3.     To enhance clarity in Figures 4, 5, 6, 7, 8, 9 and 10, the authors should label each image with an index (e.g., 'a,' 'b,' 'c,' 'd'). The current labeling method hinders clarity, as it does not clearly differentiate individual images within each figure, making it challenging for readers to connect specific parts of the figure with the text. Adding clear, indexed labels would improve the readability and comprehension of each figure. Additionally, some images within these figures appear to be overlapping or placed too closely to each other, which further hinders visual clarity.

The manuscript needs a major revision before it can be considered for publication. The authors need to address all the recommendations and suggestions mentioned above to improve their manuscript.

Author Response

Reviewer-2

Comments and Suggestions for Authors

biomimetics-3303949

In the manuscript with the title “Effect of photolithographic biomimetic surface microstructure on wettability and droplet evaporation process” the authors investigate how biomimetic surface microstructures, created using photolithography, influence wettability and droplet evaporation processes. By experimenting with various microstructures, such as square and cylindrical micro-pillars, the authors found that square micro-pillars tend to enhance hydrophobicity more effectively. In contrast, cylindrical micro-pillars reduce the temperature difference at the liquid-vapor interface by up to 18% compared to square micro-pillars, thereby mitigating evaporative cooling, and enhance droplet evaporation rates by approximately 13% compared to surfaces with square micro-pillars. Increased roughness amplifies these effects, while substrate temperature changes strongly influence surface wetting. These insights aim to support the optimal design of biomimetic surfaces. This research advances the understanding of how biomimetic surface microstructures impact wettability and evaporation dynamics, offering valuable insights for designing optimized surfaces in fields such as biomimetics, surface engineering, and thermal management.

The manuscript has some limitations and weaknesses that need to be addressed and improved. Here are some specific recommendations and suggestions for each section of the manuscript:

Materials and Experimental setup

  1. To enhance clarity in Figure 1, the authors should add a clear label to each image with an index (e.g., 'a,' 'b,' 'c') and include a description of what each image represents within the figure caption.
  2. Response to comment: Thank the reviewer for the positive comments and valuable suggestions for improving the quality of our manuscript. Sequence numbers have been added to Figure 1, and 1a, 1b, and 1c have been described separately. The modified parts have been marked with a yellow background.

  1. While the authors reference the general strategy of using micro-pillars ([46, 47]), there is no explicit linkage between these references and the specific choices of processing parameters. Although the manuscript provides a description of the processing parameters and test conditions, it would benefit from a more explicit discussion on how these choices were influenced by prior studies or preliminary experiments.
  2. Response to comment: We supplemented the relevant content of the micro-pillar structure design to make our experimental design more complete. In this study, the size of the photoresist bionic microstructure was selected based on the processing accuracy of the SU-8 material. This description has also been added to the second section of the article and marked in yellow.

Results and discussion

  1. To enhance clarity in Figures 4, 5, 6, 7, 8, 9 and 10, the authors should label each image with an index (e.g., 'a,' 'b,' 'c,' 'd'). The current labeling method hinders clarity, as it does not clearly differentiate individual images within each figure, making it challenging for readers to connect specific parts of the figure with the text. Adding clear, indexed labels would improve the readability and comprehension of each figure. Additionally, some images within these figures appear to be overlapping or placed too closely to each other, which further hinders visual clarity.
  2. Response to comment: Thank the reviewer for valuable comments on our article. Following your suggestions, we have adjusted some data graph letter numbers and corrected several mistakes in our previous draft. Based on your comments, we also adjusted the layout of the images in the paper to make them easier for readers to read and compare. All corrections are marked with a yellow background.

Round 2

Reviewer 2 Report

Comments and Suggestions for Authors

The manuscript can be accepted in its current form.